# Effects of the Developing and Using a Model to Predict Dengue Risk Villages Based on Subdistrict Administrative Organization in Southern Thailand

**DOI:** 10.3390/ijerph191911989

**Published:** 2022-09-22

**Authors:** Orratai Nontapet, Jiraporn Jaroenpool, Sarunya Maneerattanasa, Supaporn Thongchan, Chumpron Ponprasert, Patthanasak Khammaneechan, Cua Ngoc Le, Nirachon Chutipattana, Charuai Suwanbamrung

**Affiliations:** 1Excellent Center for Dengue and Community Public Health (EC for DACH), School of Nursing, Walailak University, Thasala District, Nakhon Si Thammarat 80160, Thailand; 2Excellent Center for Dengue and Community Public Health (EC for DACH), School of Allied Health Sciences, Walailak University, Thasala District, Nakhon Si Thammarat 80160, Thailand; 3School of Public Health, Walailak University, Thasala District, Nakhon Si Thammarat 80160, Thailand; 4Excellent Center for Dengue and Community Public Health (EC for DACH), The Center for Digital Technology, Walailak University, Thasala District, Nakhon Si Thammarat 80160, Thailand; 5Public Health Official of Lansaka District, Nakhon Si Thammarat 80160, Thailand; 6Excellent Center for Dengue and Community Public Health (EC for DACH), School of Public Health, Walailak University, Thasala District, Nakhon Si Thammarat 80160, Thailand

**Keywords:** dengue risk, village, larval indices, surveillance system, community participatory action research, subdistrict administrative organization

## Abstract

The purpose of this study was to evaluate the effects of developing and using a model to predict dengue risk in villages and of a larval indices surveillance system for 2372 households in 10 Thai villages. A community participatory action research method was used in five steps: (1) community preparation covering all stakeholders, (2) assessment of the understanding of a dengue solution and a larval indices surveillance system, (3) development of a prediction and intervention model for dengue risk villages, (4) implementation of the model that responds to all stakeholders, and (5) evaluation of the effects of using the model. The questionnaires to assess and evaluate were validated and reliability tested. The chi-square test and Fisher’s exact test were used to analyze the quantitative data collected by means of questionnaires. Thematic analysis was applied to the qualitative data collected through interviews. The results found that the model consisted of six main activities, including (1) setting team leader responsibility, (2) situation assessment, (3) prediction of the dengue risk in villages, (4) the six steps of the larval indices surveillance system, (5) the understanding of the dengue solution and the understanding of the larval indices surveillance system training program, and (6) local wisdom innovation. The effects of using the model showed a statistically significant increase in correct understanding among 932 family leaders, 109 village health volunteers, and 59 student leaders regarding dengue prevention and control (*p* < 0.05). The larval indices and dengue morbidity were diminished and related to the nine themes present in the community leaders’ reflections and to the satisfaction of the community members. Hence, local administrative organizations should use community-based approaches as the subdistrict dengue solution innovation to reduce the dengue problem.

## 1. Introduction

Dengue is a critical health problem in tropical and subtropical regions, especially in Southeast Asia [1,2], and has been a serious public health problem with an unpredictable outbreak pattern for more than 60 years in Thailand [3]. This is because the disease can bring about psychosocial and behavioral effects. There is an increasing comprehensive coverage of integrated interventions and services that can help target populations and beneficiaries [4]. The Thai Ministry of Public Health (MoPH), the provincial public health offices, and the subdistrict administrative organization (SAO) are responsible for the surveillance and control of the disease at the national, provincial, and local levels, respectively. In particular, the SAO executes fiscal policies and plans, including funded projects on dengue prevention and control. The legislation on disease prevention and control comprises the Public Health Act, B.E. 2535 (1992), the Subdistrict Council and Subdistrict Administrative Organization Act, B.E. 2537 (1994), and Amendment No. 6 of B.E. 2552 (2009), which apply to all SAOs [4,5,6]. However, within five years (2015 to 2019), the dengue outbreaks in the southern region of Thailand showed morbidity rates (cases per 100,000 population) of 121, 185, 136, 141, and 173, respectively, and mortality rates of 0.13%, 0.26%, 0.29%, 0.18%, and 0.32%, respectively [7].

The SAOs may be able to tackle dengue and reduce mosquito breeding grounds, but the research also found that they need to integrate other strategies for implementing the ordinances [5,6]. One study found that enforcing the Public Health Act, B.E. 2535 (1992) on four SAOs and using community participatory action research (CPAR) allowed for the implementation of local ordinances [6]. Another study explored the success and failure factors of enforcing the Public Health Act, B.E. 2535. It showed that few SAOs have an understanding of the act, which results in inadequate enforcement [5]. In addition to the participation of the community members in dengue prevention, control activities were conducted by researchers, with supporting funds from outside the community [8,9,10,11,12,13,14]. Furthermore, the knowledge, attitude, practice, and behavior of the stakeholders were found to be crucial for dengue control [15]. It is still not clear how to organize community participation to be effective in the prevention of epidemics, including that of dengue by the SAOs. 

These results denote that the SAOs should be engaged in developing and implementing activities to achieve community-based solutions for dengue [16,17]. This assumption was confirmed in a study by Alvarado-Castro et al. [18]. Traditionally, larval indices surveillance systems included the house index (HI; rate of houses infested with larvae; HI < 10), the Breteau Index (BI; number of positive containers per 100 houses inspected; BI < 50), the container index (CI; rate of water-holding containers infested with larvae; CI < 1), and the morbidity rate [19,20,21]. Studies have shown that the larval index surveys—applied to evaluate the capacity of the community in dengue prevention [22]—are feasible, low-cost, and convenient. The research suggests that dengue risk prediction should consist of village activities covering households and schools, along with LISS, the latter being an innovative solution for dengue outbreaks [23]. A literature review indicated that community capacity building was an important component of a sustainable dengue prevention solution, along with collaboration, institutionalization, and the routinization of activities [24]. The focus is on how to prevent the failure of the dengue epidemic control in the community, which has occurred due to little attention being paid to the translation from policy to health practice and community participation and to the development of disease control programs.

Then, by the SAO model, the local health teams implemented interventions, such as health education for stakeholders to better understand dengue and to develop distinctive and cutting-edge strategies for preventing mosquito bites; these comprised ten villages, three temples, five schools, four child development centers, and two primary care units, which were representative of 2372 households. There were 134 village health volunteers (VHVs). The dengue mortality rates over the preceding five years (2016–2020) in the Keawsan subdistrict, Nabon district, Nakhon Si Thammarat province, Thailand, exceeded the thresholds set by the Thai Ministry of Public Health (dengue morbidity rate of 50 cases/100,000 people and mortality rate of 0.2 percent of 100 dengue patients) [20]. 

To ensure that novelty lies in its innovative approach, though it builds on a similar study design, this subdistrict administrative organization needs a new strategy as a Keawsan SAO dengue model. As a result, the purpose of this study is to complete the steps of CPAR, which are to prepare, assess, develop, and implement an interventional model based on community participation and to evaluate how well it works in tackling dengue epidemics. 

## 2. Materials and Methods 

### 2.1. Study Setting and Participants

The Keawsan subdistrict is an SAO in Nabon district, Nakhon Si Thammarat province, Thailand, comprising 10 villages, three temples, five schools, four child development centers, and two primary care units. At the time of research, the SAO covered 2372 households, 7068 people, and 134 village health volunteers (VHVs) [2]. The SAO did not only follow legislation regarding disease prevention and control; in terms of the Public Health Act (1992), it established a decentralized plan and procedures for the prevention and control of communicable diseases [4], focusing on the community participation of all stakeholders. 

Participant size followed each step of CPAR. For the type of stakeholders, purposive sampling technique was utilized. The calculations for 687 families were performed with the G*Power 3.1 calculus program, where α = 0.05, power = 0.95, effect size = 0.15, and degrees of freedom (Df) = 1 (Df = [r − 1][c − 1]); the selected difference between the two independent means (two groups) (http://www.gpower.hhu.de/en.html, accessed on 12 June 2019) was increased by 20% to account for lost samples. Then, the sample size was 824 family leaders of a total population of 2372 families. Simple random sampling was used to select a family leader from each family, with a VHV sampling of 5 family leaders.

The purposive technique for all the VHVs and student leaders was based on the inclusion criteria; they were key stakeholders or persons with key information regarding dengue solutions and a willingness to participate. The participants were enrolled and informed of the research objectives; their informed written or oral consent was collected before they participated. In particular, consent from the students’ parents was obtained to permit their children to participate in the project (Table 1).

### 2.2. Development Step

To develop the Keawsan SAO dengue model that was appropriate for the context, an applied community participatory action research (CPAR) design was used, comprising five steps: (1) preparation; (2) assessment; (3) development; (4) implementation; and (5) evaluation.

#### 2.2.1. Preparation Step

The research team coordinated with the leader and committee of the Keawsan SAO in preparation for this study. The project plan meeting was set to be conducted with the SAO committee, which comprised 59 key leaders. Following the administrative management of the subdistrict, those available to participate comprised 10 community leaders, 20 VHVs, 2 public health providers from two primary care units, 2 from the direct public health office, 5 teachers from five schools in the subdistrict, 4 staff from four child development centers, 3 representatives from three temples, 10 members of the SAO council committee, and 3 SAO officials. The aim of the meeting was to describe the objectives of the project and brainstorm strategies to curb dengue and to provide the program timeline and the roles of all the stakeholders. It specifically aimed at preparing the data: for the computerized analytical procedures for supporting LISS; for predicting the dengue risk villages in the previous five years (the data comprised the number of dengue cases in the village, the village population, the village area, and activities for curbing dengue in the village); and for developing the computer program (https://nakhonsi.denguelim.com). A workshop was also planned and was aimed at allowing the same 59 key leaders to collect data on the larval indices surveillance and the dengue risk villages. Moreover, they were assigned their roles for participation in the model.

#### 2.2.2. Assessment Step

In order to develop the Keawsan SAO dengue model, the step consisted of assessing the housing environment across 10 villages and the application of the UDS and the ULISS questionnaires for the VHVs, family leaders, and students.

(1)Houses’ Environment Assessment

The environment of the houses in the 10 villages within the SAO’s area was assessed to set the solution plan and provide feedback data to the community. The survey format and items included participant characteristics, the environment of the houses, the water containers, and a larval index survey in eight types of containers which might be mosquito breeding sites [19,20]. According to the sample size of larval indices (house index), an estimated 300 houses were inspected per 10,000 houses [21]; therefore, the subdistrict needed at least 300 houses from a total of 2372 households. Selecting the sample houses was by a simple random sampling of 3 houses from a total of 10–15 houses that responded with a VHV. However, the data collection course was by the training of all the VHVs by the researcher. 

(2)UDS and ULISS Assessment

In the six-step cognitive domain of Bloom’s revised taxonomy for learning, “understanding” consists of six cognitive levels: knowledge, understanding (comprehension), application, analysis, synthesis, and evaluation [25,26]. In this study, the understanding of the dengue solution (UDS) refers to the capacity of the VHVs to comprehend dengue prevention, control, and self-care. Meanwhile, understanding the larval indices surveillance system (ULISS) refers to the capacity of the VHVs to comprehend the larval indices that characterize the larval indices surveillance system processes and the larval index levels. To resolve the dengue-related issues in Thailand, the VHVs surveyed the larval indices in certain areas once a week or once a month and thereafter communicated the dengue risk to the community members. 

#### 2.2.3. Development Step

(1)Predicting High- and Low-Risk Dengue Villages

The risk predictions were stratified as high-risk dengue village (H-RDV) or low-risk dengue village (L-RDV); to predict the risk of dengue in the two primary care units of the 10 villages of the subdistrict, we used a CPAR procedure, which was divided into the following three steps;

Step 1. Determine the dengue risk using the dengue risk assessment criteria and the prediction of high/low risk criteria from the study in southern Thailand [12]. Dengue risk refers to an opportunity for dengue to emerge in a region. The following factors related to the emergence of dengue in the previous five years were considered: 

The dengue severity aspect comprised three factors: (1) the endemic village factor, referring to the ongoing incidence of dengue cases; (2) the dengue herd immunity factor, which was demonstrable within the community and defined by the dengue morbidity rate of each village; and (3) the current morbidity rate factor, referring to the value comparing the current dengue morbidity rates with the median rates of the past five years. 

The dengue outbreak opportunity aspect comprised three factors: (1) the population movement, which refers to whether the area facilitates migration and allows for the virus to circulate in specific regions; (2) the population density in the village, which is self-explanatory and is an integral element considering that dengue spreads through mosquitoes (i.e., a dense population could increase the risk of a dengue epidemic); and (3) the intensifying of the factors in villages for dengue prevention operations, including activities, projects, or interventions for dengue prevention in the villages. These interventions should be characterized as assessments of the dengue risk criteria, generally referring to the following five activities: the larval indices surveillance system, waste management, the larval indices level of the village, community capacity-related operations, and school-based dengue prevention operations. Together, the dengue severity aspect and the dengue outbreak opportunity aspect comprised 33 criteria, having a total score of 33 points.

Step 2. A village-level evaluation of the risk areas that were under the responsibility of a primary care unit was conducted. For villages that had two primary care units, the evaluation was conducted according to the specified criteria established during a conference, wherein official representatives were tasked with solving the dengue-related problems on behalf of each primary care unit. Furthermore, the data on the illness rate, mortality rate, and total population of each village were comprehensively analyzed.

Step 3. Villages with H-RDV and L-RDV were defined using half of the total scores (i.e., 33 points, 15 points from the dengue severity aspect and 18 from the dengue outbreak opportunity aspect). The risk cut-off value was set at 17 points. If a village scored 17 or more points, it was an H-RDV; if less than 17, an L-RDV, based on the two levels of risk that were confirmed in a previous study; this can be practically understood as high and low [12]. (Appendix A: Keawsan SAO model: the computer program for dengue risk village prediction).

(2)Setting Larval Indices Surveillance System

The larval indices surveillance system of the Keawsan SAO model applied the steps of the model [10]. In total, 134 village health volunteers from 10 villages in the subdistrict participated in the workshop-based UDS and ULISS training programs. The 134 village health volunteers (VHVs) were divided into 3 to 4 zones per village and 5 to 6 VHVs per zone. The two primary care units were at the center of the surveillance system of the subdistrict. 

(3)The UDS and ULISS Training Program

The program helped to predict the dengue risk villages and in the setting up of the larval indices surveillance system. The training programs focused on the dengue solution content and the larval indices surveillance system for the key stakeholders, including community leaders, village health volunteers (VHV), and student leaders.

(4)Develop the Local Wisdom Innovation (LWI) with Herbs for Mosquito Repellant

The Keawsan SAO model team set up a plan for tackling dengue through interventions, such as predicting the risk of dengue in the 10 villages, implementing education programs for the stakeholders to better understand dengue and the larval indices surveillance system, and creating unique and innovative mosquito bite preventions. Based on budgetary support from the SAO, there were four steps for developing the local wisdom innovation of the 10 villages and 4 child development centers: (1) preparation with a workshop about producing the local wisdom innovation products; (2) The meeting conducted by the village leader group and the child development center; (3) the testing of the method for use; and (4) evaluation of the method for use in the community.

#### 2.2.4. Implementation Step

We concluded that all the activities of the model were conducted for 12 weeks. The activities comprised the UDS and ULISS training programs and the designation of responsibilities for each stakeholder. 

#### 2.2.5. Evaluation Step

This step was the evaluation of the process of development and the effects after using the Keawsan SAO dengue model. This step measured and compared the UDS and ULISS of the family leaders, VHVs, and student leaders before and after using the model. Moreover, it evaluated the larval indices level, the dengue morbidity rate, and the satisfaction of the stakeholders from the meetings of all the stakeholders.

### 2.3. Questionnaires for Assessment and Evaluation

These self-reported questionnaires for UDS and ULISS comprised three parts and were applied to the VHVs. Part I included eight items on general characteristics such as sex, age, education level, experience, and duration of VHV role. Part II comprised 15 items that assessed the VHVs’ UDS. Part III assessed the ULISS and also consisted of 15 items. This survey took between 20 and 30 min to complete. 

The questionnaires for UDS and ULISS were administered as described in previous research [27]. Three experts assessed the content validity indices (CVIs) of these questionnaires, yielding values of 0.90 for the UDS and 0.91 for the ULISS; these values denoted that both the questionnaires were valid. Regarding reliability, this refers to the degree to which the results obtained by a measurement and procedure can be replicated. A preliminary study with 120 VHVs to analyze the reliability returned Cronbach alpha coefficients of 0.70 and 0.75 for the UDS and ULISS, respectively. A reliability coefficient (alpha) of 0.70 or higher was considered to represent acceptable reliability [28].

The UDS and ULISS of the family leaders and student leaders were assessed using the same questionnaires, modified to 10 UDS items and 10 ULISS items. Three experts assessed the CVIs, yielding values of 0.82 for the UDS items and 0.88 for the ULISS items, which verified their validity. Reliability testing resulted in Cronbach alpha coefficients for the UDS and ULISS of the 60 family leaders of 0.80 and 0.81, respectively, and for the 30 students, the Cronbach alpha coefficients for the UDS and ULISS were 0.72 and 0.73, respectively; these values showed acceptable reliability (alpha) of 0.70 or higher [28]. These questionnaires also comprised three parts similar to those described above.

Based on Bloom’s cut-off score of 80% [26,29], we devised mean scores for both the UDS and the ULISS to describe good and poor levels of understanding. Specifically, obtaining 80% or more than 80% of the answers correct indicated a good understanding, whereas getting less than 80% indicated a poor understanding. Accordingly, good levels of understanding were denoted by the VHVs who scored 12 points or more and the households and students that scored 8 points or more on the UDS and ULISS questionnaires.

The questionnaire for measuring the project satisfaction was a short, six-item questionnaire that assessed opinions of overall satisfaction. Three experts reported CVI values of 0.92. The semi-structured questionnaires did not require reliability verification. The items were rated on a five-level scale of opinion: strong agreement, agreement, neutral, disagreement, and strong disagreement. 

At the evaluation step, the stakeholders met for discussion and reflection. The questionnaires for discussion by the community leaders and students were set based on unstructured questions. The participants reflected on aspects of the project, such as participation in the activities of the village, the larval indices surveillance system, the dengue training program, and the project’s utilities and barriers, and they made suggestions for solving dengue.

### 2.4. Data Analysis

The data that served to predict the risk of dengue in a village were analyzed using the entry assessment criteria, and the ULISS survey data were analyzed using a computer program (https://nakhonsi.denguelime.com). The larval indices (BI, HI, and CI) were analyzed by the ratio, frequency, and percentage of the type of infestation found in the water containers. The data from the house environment survey, the six items that measured the stakeholders’ satisfaction and the participants’ characteristics were collected from the 134 VHVs and analyzed using descriptive statistics. The chi-square test and Fisher’s exact test were used to compare the scores for the UDS and ULISS of the VHVs, family leaders, and student leaders at the beginning point and the endpoint of the training program, and the results of the LISS were shown in the figures. 

The qualitative data from the group discussion meetings of the VHVs and stakeholders were analyzed by applying Braun and Clarke’s thematic analysis [30,31]. Two researchers conducted and analyzed the community leaders’ reflections that were provided in meetings conducted during the closing of the project.

## 3. Results

The effects of study are (1) the model that was named the “Keawsan SAO dengue model”, which consisted of the six main activities, and (2) the outcomes of using the model. 

### 3.1. Keawsan SAO Dengue Model

The six main activities of the Keawsan SAO dengue model were the setting of the team leader’s responsibility, the situation assessment, the prediction of the dengue risk villages, the larval indices surveillance system, the UDS and ULISS training program, and the local wisdom innovation.

#### 3.1.1. Setting Team Leader’s Responsibility

The preparation step showed that all the stakeholders in the Keawsan SAO dengue model had their responsibilities defined as the team leaders of the participation (Table 2).

#### 3.1.2. Situation Assessment

(1)Houses’ Environment Assessment

This section shows the assessment results for the 2372 housing environments and for the first larval indices surveillance system survey in June 2020. Of the respondents, 793 participants represented 793 households of the total houses (33.4% and of the sample size 793/824; 96.24%). Most were women (522/793; 65.8%), aged between 36 and 59 years (450/793; 56.7%), followed by Buddhists (743/793; 93.7%), those who had a non-leader status in the community (730/793; 92.1%), and those who had higher elementary education (426/793; 53.7%). Most participants gave farmer as their main occupation (504/793; 63.7%); approximately one-fifth had experienced dengue illness (165/793; 20.4%); and slightly more than one-tenth had experienced dengue illness in the neighborhood (93/793; 11.7%). 

The average family income was THB 10,645 per month (SD. = 14,480); on average, the residents had resided in the community for 36.57 years (SD. = 19.6), had four family members, and received formal dengue training 0.34 times per year (SD. = 0.47). Most of the 719 houses were situated in a rural area (90.7%); 664 were a single house (83.7%); and 627 had a single story (79.1%). 

We observed that only 184 (23.2%) households had a tidy environment. Regarding water resources, usable water was piped to almost all of the 793 houses (88.5%); drinking water was supplied by water tanks to 679 households (85.6%); and 747 households had a toilet in the house (94.2%). The total numbers of water containers surrounding or within the houses were 3048 and 2684, respectively, and the average ratio was 3 to 4 containers per house. Moreover, there were 749 unused containers surrounding the houses.

#### 3.1.3. Village Dengue Risk Prediction Criteria 

The 10 villages were stratified into five H-RDVs (50%) and five L-RDVs (50%). The six villages (villages 1, 2, 5, 7, 9, and 10) for which PCU_1_ was responsible showed L-RDV, whereas the villages numbered 3, 4, and 8, which had PCU_2_ responsible of their health status, were H-RDV (Table 3).

#### 3.1.4. The Larval Indices Surveillance System 

The larval indices surveillance system is based on the following six steps: (1)In total, 2372 family leaders were surveyed using the larval indices and were helped in the elimination of the breeding sites of mosquitos in their houses and within the surrounding 100 meters of their houses every seven days.(2)The 134 VHVs and family leaders underwent surveys on the larval indices of 10 to 15 households, 5 schools, 4 child development centers, and 4 four temples. Their data were inserted in their “violet books”—notebooks for recording the results of the household larval data from the 10 to 15 houses surveyed per a VHV on 25th of each month—and the books were sent to the 30 zone leaders.(3)The 30 zone leaders collected the survey data from their VHVs, inserted them in their “blue books”, and sent the books to the community leader on the 28th of each month.(4)The 10 community leaders collected the data from each zone leader, inserted them in their “yellow books” on the 30th of each month, and sent those to the two primary care units.(5)Two primary care units gathered and entered the data from all villages into the website http://nakhonsi.denguelim.com. The levels of the larval indices (BI, HI, and CI) were reported in a VHV meeting on the 9th of each next month. All the stakeholders were then used to construct a local wisdom innovation.(6)The primary care units communicated the findings to all the stakeholders, to the SAO, and to the district public health official.

The six steps of the larval indices surveillance system were followed step-by-step, going from the households to the two primary care units and to the SAO (Figure 1).

#### 3.1.5. The UDS and ULISS Training Programs

(1)The UDS and ULISS training programs were applied to 50 stakeholders in the community and the SAO. At the program’s onset, the UDS training was applied; the SAO devised the training for all the stakeholders regarding the objectives of the methods to tackle dengue, as well as the meaning, causes, signs, symptoms, and treatment for dengue illness. Thereafter, the ULISS training was applied, focusing on explaining the seven steps of the LISS and the roles of the stakeholders who participated in and supported the initiative.(2)The UDS and ULISS training programs were applied to all 134 participating VHVs; this is because the VHVs were the key stakeholders for tackling dengue in villages where the UDS and ULISS training programs were conducted. They underwent workshops for six months and completed the 15-item UDS and ULISS questionnaires pre- and post-training (in June and August 2020). After the village health volunteers’ training, they were requested to share their understanding of dengue solutions and the larval indices surveillance system with the family leader during the larval indices survey on every 25th day of each month from June to August 2020.(3)The UDS and ULISS training programs were also applied to 59 student leaders who represented five schools. They also completed the questionnaires before the test, in June 2020, and post-test, after training for 3 months.

#### 3.1.6. Local Wisdom Innovation

According to the village dengue risk score and the context of the village, the VHVs, community leaders, child development centers, and family leaders were encouraged to create the local wisdom innovation in villages 1–10, such as (1) garbage-free communities, with three colored flags for community-based methods to avoid dengue and the aroma of citronella as a mosquito repellent, (2) placing reminder stickers for eradicating *Aedes aegypti* mosquito larvae, (3) herbal candles as a mosquito repellent, (4) herbal spray as a mosquito repellent, (5) a herbal scented bag as a mosquito repellent, (6) citronella essential oil for dengue elimination, (7) conducting community and household environment management and elimination of *Aedes aegypti* mosquito larvae, (8) red lime (Ca(OH)_2_) for reducing the *Aedes aegypti* mosquito larvae, (9) citronella as a mosquito repellent, and (10) volunteers to create a network for dengue prevention. 

### 3.2. Effects after Using Keawsan SAO Dengue Model

#### 3.2.1. VHVs’ UDS and ULISS 

These were conducted at the beginning of the project in the SAO. At the onset of the UDS and ULISS training programs, there were 109 VHVs; at the end, there were 90. There were no significant differences between before and after the implementation of the training programs, except for the duration of the training program of 12 weeks per year (*p* < 0.05). 

A comparison of the VHVs’ scores for the UDS before and after implementation showed that 13 items (out of 15 items) had statistically significant differences (*p* < 0.001), and one item (item number 12) was less significantly different (*p* < 0.01). Meanwhile, the scores for item 13 were not significantly different (*p* > 0.05; Table 4).

Regarding the comparison of the VHVs’ scores for ULISS before and after implementation, they showed an increase in the ratio of scores with good levels, and fourteen items showed an increase in the correct response ratio that was statistically significant (*p* < 0.001); meanwhile, a single item did not show a statistically significant increase for correct response ratios (*p* < 0.05; Table 5).

#### 3.2.2. Family Leaders’ UDS and ULISS 

Family leaders received information from the VHVs who were trained in UDS and ULISS activities. On the 25th of each month, the solutions to the dengue and larval indices surveillance system were communicated to households by the VHVs. All villages participated in the local wisdom innovation activities. The representative households participated for 12 weeks, and they were accompanied by 109 and 90 VHVs before and after implementation, respectively. The numbers of family leaders before and after the implementation of the intervention were 932 and 856 individuals, respectively. There were no significant differences between the scores before and after the implementation for female family leaders (50.9%: 49.1%), those aged below 60 years (50.8%: 49.2%), and those living in the community for more than 10 years (51.8%: 48.2%; *p* > 0.05). Nonetheless, four personal characteristics showed significant differences (*p* < 0.05): having senior high school education (57.6%: 42.4%), dengue training time of more than twice per year (48.6%: 51.4%), no experience of dengue in the previous year (50.8%: 49.2%), and survey and elimination of larvae breeding sites every seven days (47.2%: 52.8%).The results of the comparison of the family leaders’ scores for the UDS before and after implementation showed an increase in good level score ratios and correct response ratios, which were statistically significant (*p* < 0.001; Table 6). 

Regarding the scores for the ULISS, there was also an increase in the good level score ratios and the correct response ratios, which were statistically significant (*p* < 0.001; Table 7).

#### 3.2.3. Student Leaders’ UDS and ULISS

In total, 59 student leaders, who represented five schools, participated in a single group that underwent the UDS and LISS training programs. The majority were girls (59.5%), aged 10 or older (91.5%), in junior elementary school (79.9%), completed larval indices surveys once a month (47.5%), had lived in the subdistrict area for more than 10 years (79.7%), had dengue training for more than one year (52.5%), and had no experience of dengue illness (76.3%). Only a few students had experienced dengue illness in their neighborhood (5.1%) or in their family (8.5%).

Regarding student leader scores for the UDS and the comparisons between results before and after the implementation, we observed that five items showed statistically significant differences (*p* < 0.001; an item was statistically different if *p* < 0.01). The scores for item 5 were statistically significantly different (*p* < 0.05), and the scores for items 1, 2, and 4 showed no significant differences between the implementation periods (*p* > 0.05; Table 8).

Regarding the student leaders’ scores for the ULISS, seven items showed an increase in the correct response ratios that were statistically significant (*p* < 0.001; Table 9).

#### 3.2.4. Larval Indices Level before and after Using the Model

According to the Keawsan dengue project, the SAO should implement measures to predict the dengue risk villages and implement the larval indices surveillance system and the UDS and ULISS training programs for community leaders, VHVs, family leaders, and students and conduct house environment surveys. Moreover, all stakeholders were encouraged to use the provided data for developing local wisdom innovation in the 10 participating villages and 4 participating child development centers to decrease the larval indices. The full results for the Breteau index (BI), house index (HI), and container index (CI) before and after implementation are shown at https://nakhonsi.denguelim.com (accessed on 8 October 2020). The larval indices of all 10 villages decreased after implementation. Thereafter, we analyzed the data that showed the effects post-implementation, such as the scores, the number of stakeholders, the larval indices before and after implementation, the names of the local wisdom innovations, and the larval indices (Table 10).

#### 3.2.5. Dengue Morbidity Rate

The preparation, assessment, development, implementation, and evaluation of the Keawsan SAO dengue model took place during the last six months of 2020. According to the model, the prevention and control of dengue needs all the stakeholders in the subdistrict. Thereafter, the dengue morbidity rate in 2021 (152 cases per 100,000 population) decreased compared with the data from the previous five years (201 cases in 2016, 218 cases in 2017, 267 cases in 2018, 350 cases in 2019, and 533 cases in 2020; Figure 2).

#### 3.2.6. Satisfaction of Stakeholders

In total, 234 stakeholders were females (67.9%), aged between 35 and 59 years (65.4%); they received UDS and ULISS training from the VHVs (64.5%), engaged in activities for eliminating larvae in and out of their houses every seven days (77.4%), participated in the documenting activities of VHVs on the 25th of each month (67.1%), and participated in the local wisdom innovation of the village (54.3%) and in the school and the child development center (30.3%). The satisfaction levels of the participants in the Keawsan SAO dengue model were measured as part of the evaluation component of the community participatory action research. Almost all the stakeholders were highly satisfied: (1) the model decreased the dengue morbidity rate (85.0%); (2) the stakeholders had to participate in the dengue solution activities (87.6%); (3) the local wisdom innovation of the village could decrease dengue severity (84.2%); (4) the success of the model needed family leaders to eliminate mosquito larvae (89.4%); (5) the community needed to continue conducting projects to eliminate dengue (91.5%); and (6) the overall satisfaction of all the stakeholders was high (85.9%).

In the final project meeting, 52 individuals represented the stakeholders, including SAO officials, teachers, VHVs, community leaders, and students. They discussed the solutions, processes, and effects of the program, which comprised all the activities mentioned above. Using thematic analysis, we summarized their reflections and extracted nine themes: (1) every person in the community must participate in the dengue solution initiatives; (2) members of the households must know and practice methods to tackle dengue; (3) villages should participate in dengue solution initiatives; (4) the VHVs must be responsible for and show the best ways to practice the methods to tackle dengue; (5) the community leaders must communicate and coordinate with other parties; (6) the SAO must participate in the initiatives and assign budgets; (7) the PCUs should serve as academic mentors; (8) the schools must be aware of the methods for curbing the spread of dengue; and (9) the Keawsan SAO dengue model must be strengthened by stakeholder empowerment. 

## 4. Discussion

In this CPAR, we followed five steps to prepare, assess, develop, implement, and evaluate the effects of the Keawsan SAO dengue model, which was meant to provide a sustainable model for dengue prevention and control based on the SAO. We used the concepts of CPAR to evaluate the village dengue risk and the larval indices surveillance system; these methods concur with the methods proposed by previous research using the larval indices surveillance system within a district and the high- and low-dengue-risk primary care units [11]. However, these studies did not analyze SAO support in the implementation of the methods to curb dengue. In this study, the SAO played a role in supporting, coordinating, charting, participating, and continuing dengue prevention initiatives using local ordinances according to the Public Health Act, B.E. 2535 (1992). It focused on producing community participation, such as by involving community leaders, family leaders, village health volunteers, students, and primary care units, thereby helping the local governmental administration to achieve vector control [5,32].

Based on the previous research, in the preparation step the stakeholders appointed a team leader for planning the projects, prepared data for the prediction of the dengue risk villages using a computer program (http://nakhonsi.denguelim.com) (accessed 8 October 2020), developed the guidelines for community participation, and developed and implemented training programs to build community capacity in the area [11]. The assessment of the situation of the Keawsan SAO revealed a high risk of dengue outbreaks, which were described by: (1) high dengue morbidity rates in the previous five years; (2) high larval indices levels (i.e., BI, HI, and CI; Table 9); (3) inappropriate characteristics of the houses in the area for dengue prevention and control; and (4) numerous incorrect responses from stakeholders on the understanding of the dengue solution (UDS) and the understanding of the larval indices surveillance system (ULISS) questionnaires before the implementation of the training programs. These factors were consistent with those found in a study of population density, water supply, and dengue risk in Vietnam. It showed areas with a high population density or an adequate water supply without severe dengue outbreaks; meanwhile, the risk of dengue was higher in rural areas than in urban areas, which was largely explained by the lack of a supply of tap water and the decrease in local population densities being more frequently in the endangered range [33]. However, before implementation, the incorrect responses of stakeholders to the UDS and ULISS questionnaires indicated a serious lack of capacity building associated with knowledge, attitude, and practice of dengue prevention and control measures; these findings concur with two previous studies [17,34].

Developing and using the Keawsan SAO dengue model in 10 villages and 4 child development centers to ensure the controlling of dengue comprised the following steps: (1) setting team leader responsibility, (2) situation assessment, (3) prediction of village dengue risk, (4) the six steps of the larval indices surveillance system, (5) the understanding of the dengue solution and the understanding of the larval indices surveillance system training program, and (6) local wisdom innovation with herbal mixtures as mosquito repellents. To ensure the transferring of knowledge to village residents, all stakeholders in the dengue prevention in the SAO (i.e., VHVs, community leaders, public health officials of the PCUs, and SAO officials) participated in the UDS and ULISS training programs. They are the key stakeholders in dengue prevention; this is similar to the previous community-based studies that placed importance on the participation process for dengue prevention and control [3,10,13,19,21]. In particular, a study described the SAO official need to understand how to tackle dengue when implementing the local provision of law ordinances enforced by the Public Health Act of 1992 [4]. Following in the footsteps of previous research [12], after predicting the H-RDV and L-RDV, we set the LISS, which involved all stakeholders through the processes present in the six steps and other activities. The implementation spanned three months, and our findings showed that, post-implementation, it helped to develop activities to tackle dengue, decreased larval indices (BI, HI, and CI), and led to the lowest mortality rates in the previous five years [12]. We believe that model owes it effective implementation to its community-based intervention, which covered all stakeholders; in this sense, it was similar to the Lansaka model and the implementation of the LISS in high- and low-risk PCUs.

The ideas underlying our methods (e.g., predicting the RDV and using community-based operations to enhance the community’s capacity to tackle dengue) are consistent with the concepts of integrated vector management advocated by the World Health Organization [1]. Moreover, a previous study showed that the prediction results helped strengthen the community in conducting dengue prevention operations in H-RDVs and L-RDVs in the subdistrict [11], as did a study in 220 villages in 24 subdistricts [12]. Additionally, the differences regarding the village dengue risks between the 10 villages were assessed based on direct contact with each village, and the two PCUs were the coordinating units for each subdistrict (which cover the villages) because the SAO is the body responsible for setting up the LISS and did not use the legal provision; a similar procedure was reported in the Lansaka model [10].

These effects clearly show the results of the proposed intervention for controlling dengue, and they concur with the studies that evaluated the effectiveness of a community-based intervention for controlling dengue in Africa [13,14]. The SAO empowered the model by providing a supportive budget, which led to consistent local wisdom innovations. The current research involved a quasi-experimental study of community empowerment. As our proposed model was shown to decrease the larval indices values, it concurred with the findings of a study exploring the role of water containers between two groups in Lampung Province, Indonesia [35]. We also observed that most community members showed high satisfaction with the model and that they required continued community projects and perceived that the LWIs of the 10 villages and 4 child development centers decreased the dengue outbreaks. Moreover, the nine themes we found through thematic analysis depicted that most participants had positive reflections toward the Keawsan SAO dengue model. These positive results demonstrated the community awareness of the benefits of dengue prevention and control through the community-based interventions against the dengue model, and they are similar to those of the previous research [13,36]. However, in India, a systematic review presented inconsistent results, showing that community-based dengue control programs were ineffective there because in almost all of the articles that were reviewed the vertically structured program through fogging and larval control was found, without any involvement of community-based dengue control program [37].

## 5. Conclusions

The Keawsan SAO dengue model is a local innovation that used a community- and SAO-based approach more frequently than simply enforcing local ordinances for tackling dengue; the proposed Keawsan SAO dengue model consisted of appointing community leaders, monitoring the situation of the dengue and housing environments, preparing data for predicting the village dengue risk and for establishing a surveillance system, and developing and implementing a UDS and ULISS training program for VHVs, family leaders, students, child development centers, and community leaders. It also included the setting up of the six steps of the LISS, creating the website http://nakhonsi.denguelim.com (accessed on 8 October 2020), and designing 14 local wisdom innovations. The evaluation of the effects before and after implementation showed an increase in scores for both the UDS and the ULISS questionnaires for the VHVs, family leaders, and students. Moreover, decreasing larval indices and dengue morbidity rates were related to the nine themes that were extracted from the discussions with community leaders and to the high levels of satisfaction among community members. However, the study might have been appropriate in Thailand but may prove inappropriate elsewhere. Therefore, the implementation of the prediction of H-RDVs and L-RDVs and the larval indices surveillance systems of the villages needs the support of the SAO’s budget or local organization and the participation of all stakeholders; the cohort study is warranted in the conducting and monitoring of activities for their long-term effects on sustainable dengue prevention and control in the future research.

## Figures and Tables

**Figure 1 ijerph-19-11989-f001:**
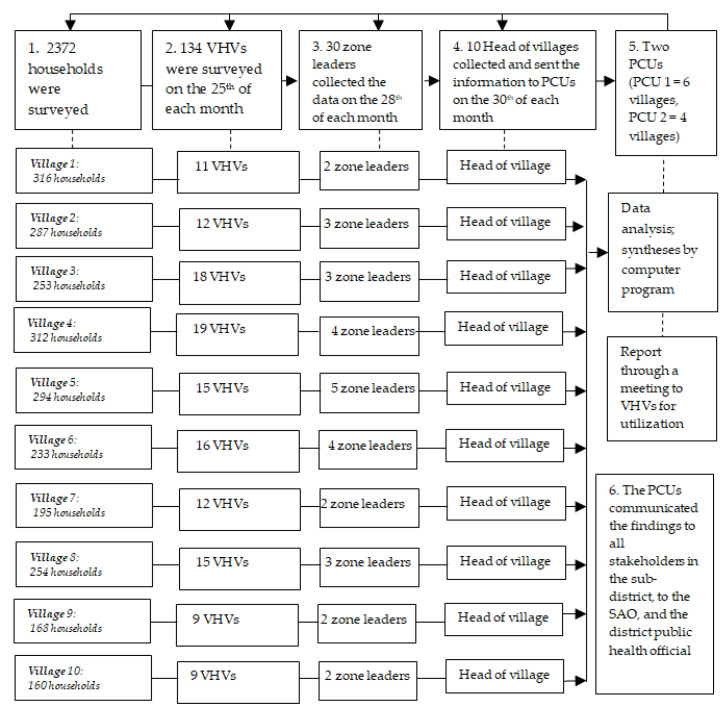
Larval indices surveillance system in 10 villages of the subdistrict. SAO = subdistrict administration; VHV = village health volunteer; PCU = primary care unit.

**Figure 2 ijerph-19-11989-f002:**
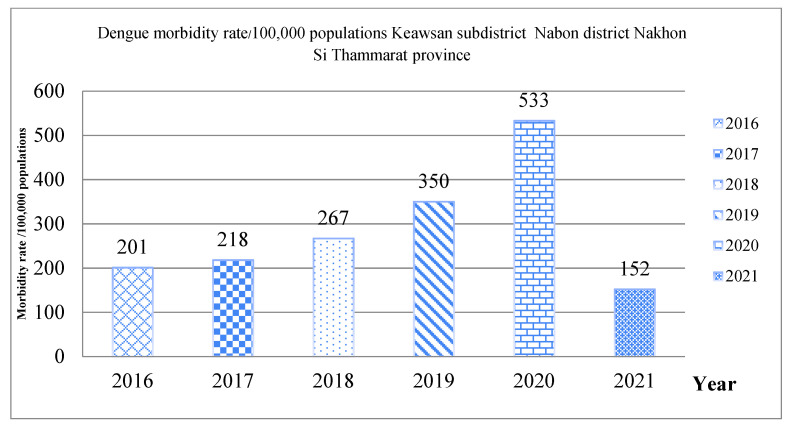
Compare dengue morbidity rate of Keawsan subdistrict.

**Table 1 ijerph-19-11989-t001:** Stakeholders and Representative of Stakeholder in the community participatory action research steps.

	Total	Stakeholder/Representative of Stakeholder of the Subdistrict									
CPAR Step		Family Leader (N = 2372)	VHV (N = 134)	Student Leader (N = 59)	Teacher of 5 Schools (N = 5)	Staff of CDC (N = 4)	Community Leader (N = 10)	Temple (N = 3)	SAO Committee (N = 10)	PCU (N = 2)	PHOD (N = 2)
Preparation step		3	20		5	4	10	3	10	2	2
Houses’ environment assessment		824 *									
Development step: LISS			134 **				10			2	
Development step: LWI			134 **			4			2	2	
Implementation step		2372	134 **		5	4	10	2	10	2	2
Evaluation: UDS, ULISS		824 *	134 **	59							

N = total stakholder; n = Representative of Stakeholder; SAO = subdistrict administrative organisation; CDC = children development center; PCU = primary care unit; VHV = villadge health voluntee; PHOD = Public health officer from the district; * the same group of Family Leader, ** the same group of VHV.

**Table 2 ijerph-19-11989-t002:** Stakeholders of the Keawsan SAO dengue model and their responsibilities for the implementation model.

Stakeholders	Responsibility
Family leaders /community leaders (N = 2372)	Responsible for conducting the survey on the household larval indices every 7 days. They were supposed to collaborate with VHVs in activities related to dengue prevention and control by learning through the UDS and ULISS training programs, which provided them with methods for achieving local wisdom innovation.
VHVs (N = 134)	Responsible for conducting the larval indices surveillance system, the assessment of the risk of dengue in the village, and monitoring community innovation. They were also supposed to communicate about the UDS and ULISS to every household via survey activities and stickers of dengue prevention guidelines for the householders.
PCUs (N = 2)	Responsible for coordinating VHVs and to enter data into http://nakhoni.denguelim.com (accessed on 8 October 2020) Responsible for communicating about dengue information and the risk level to all stakeholders, as well as for supporting dengue prevention/control activities.
SAO	Responsible for setting up plans and strategies; supporting with budget; monitoring project processes; using information to support local wisdom innovation; and promoting a continued dengue prevention/control plan.
The district public health official	Responsible for communicating to all stakeholders and supporting the SAO.
59 Student leaders in 5 schools 4 Child development centers 3 Temples	Responsible for collaborating with VHVs, PCUs, households, and the SAO for dengue prevention/control initiatives, and for providing data on larval indices through surveys at least every 7 days; they were also supposed to make good use of the initiatives of other parties to promote local wisdom innovation.

UDS = understanding dengue solution; ULISS = understanding larval indices surveillance system; PCU = primary care unit; VHV = village health volunteer; SAO = subdistrict administrative organization.

**Table 3 ijerph-19-11989-t003:** Villages with high and low risk of dengue in the Keawsan SAO.

Village Dengue Risk Prediction Criteria (RDVPC)	Point	Village									
		1 ^a^	2 ^a^	3 ^b^	4 ^b^	5 ^a^	6 ^b^	7 ^a^	8 ^b^	9 ^a^	10 ^a^
**1.** **Dengue severity aspect**											
1.1. Endemic village factor	**5**	5	2	**5**	**5**	5	**5**	3	3	3	3
1.2. Dengue herd immunity factor	**5**	2	4	**5**	**5**	1	**5**	1	5	1	1
1.3. Current morbidity rate factor	**5**	1	1	1	1	1	1	5	1	1	1
** *Total dengue severity* **	** *15* **	* **8** *	* **7** *	* **11** *	* **11** *	* **7** *	* **11** *	* **9** *	* **9** *	* **5** *	* **5** *
**2.** **Dengue outbreak opportunity aspect**											
2.1. Population movement factor	**3**	1	1	1	1	1	1	1	1	1	1
2.2. Population density in village	**5**	1	1	1	1	1	1	1	1	1	1
2.3. Strengthening village for dengue prevention activities	**10**										
(1) Larval indices surveillance system		1	1	2	2	1	2	1	2	1	1
(2) Garbage management		1	1	1	1	1	1	1	1	1	1
(3) Larval indices level of the village		1	1	1	1	1	1	1	1	1	1
(4) Community capacity activities		1	1	1	1	1	1	1	1	1	1
(5) School-based dengue prevention activities		1	1	2	2	1	2	1	2	1	1
** *Total dengue outbreak opportunity aspect* **	** *18* **	** *7* **	** *7* **	** *9* **	** *9* **	** *7* **	** *9* **	** *7* **	** *9* **	** *7* **	** *7* **
**Total full score**	**33**	**15 ***	**14 ***	**20 ****	**20 ****	**14 ***	**20 ****	**14 ***	**18 ****	**12 ***	**12 ***

RDV = risk dengue village; H-RDV = high risk of dengue in the village; L-RDV = low risk of dengue in the village; RDVPC = Risk of dengue in the village prediction criteria; PCU = primary care unit, ^a^ = village in PCU_1_, ^b^ = village in PCU_2_ * L-RDV ** H-RDV.

**Table 4 ijerph-19-11989-t004:** Comparison of correct response rates regarding the UDS for VHVs before and after implementation of the training programs.

VHVs’ UDS		Correct Response n (%)		$X^{2}$
		Before (n = 109)	After (n = 90)	
1. If a patient has a high fever for 2 to 7 days, petechiae, and a painful enlargement of the liver, the patient is showing signs of dengue infection.		94 (51.4)	89 (48.6)	10.67 ***
2. If the patient presents signs and symptoms of dengue, then the patient is showing signs of a dengue viral infection.		84 (48.6)	89 (51.4)	20.67 ***
3. If a person living in a high dengue risk area is infected with one dengue serotype, they may have lifelong immunity to that strain. However, they would still be vulnerable to other serotypes and could, thus, be infected with the other dengue serotypes later in life.		17 (16.8)	84 (83.2)	119.19 ***
4. If a patient is protected from female *Aedes aegypti* bites, they will be safe from dengue.		77 (48.4)	82 (51.6)	16.03 ***
5. If a patient has a high fever for 2 to 7 days, nausea, vomiting, and possible abdominal pain, the patient is in the fever stage.		58 (39.5)	89 (60.5)	46.15 ***
6. If a patient with dengue hemorrhagic fever presents pain at the right lower costal margin, they are showing signs of hepatomegaly.		59 (39.6)	90 (60.4)	55.14 ***
7. If a patient with dengue hemorrhagic fever presents signs of shock from leakage of plasma, they will have poor tissue perfusion, weak pulse, and narrowed pulse pressure.		61 (40.4)	90 (59.6)	52.23 ***
8. If your neighbor presents signs of poor tissue perfusion, weak pulse, and clammy skin, you need to send them to hospital.		82 (48.2)	88 (51.8)	20.13 ***
9. Dengue patients should avoid consuming aspirin or non-steroidal anti-inflammatory drugs because they may cause gastritis and subsequent massive gastrointestinal or hepatic injury.		43 (33.1)	87 (66.9)	71.25 ***
10. If your neighbor presents high fever on day 1, you give one paracetamol every 6 h and a tepid sponge bath.		74 (46.3)	86 (53.8)	23.95 ***
11. You suggest for your neighbor to prevent mosquito bites by using skin lotion.		56 (39.2)	87 (60.8)	50.01 ***
12. If a village has a dengue disease index above the threshold, you suggest a dengue prevention strategy that will destroy the mosquito breeding grounds and larva around the houses.		93 (51.7)	87 (48.3)	7.45 **
13. *Tiliacora* leaves do not serve as a natural herbal remedy for concocting a mosquito repellent.		61 (42.1)	84 (57.9)	0.06 ^ns^
14. Temephos sand granulates may be used to eliminate larval mosquitos, but not to eliminate mosquito eggs.		7 (8.0)	81 (92.0)	72.76 ***
15. Household members must complete larval surveys and eliminate contaminated water containers in their house every 7 days.		34 (54.8)	87 (45.2)	88.67 ***
**Total**	Poor level (cut-off point < 12)	103 (100.0)	0 (0.0)	0.000 ***^, a^
	Good level (cut-off point ≥ 12)	6 (6.3)	90 (93.8)	

VHV = village health volunteer; UDS = understanding dengue solution; $X^{2}$ = chi-square test. ^a^ Fisher’s exact test. ^ns^ Not significant; ** *p* < 0.01. *** *p* < 0.001.

**Table 5 ijerph-19-11989-t005:** Correct response rates regarding the ULISS for VHVs before and after implementation of the training programs.

VHVs’ ULISS		Correct Responses, n (%)		
		Before (n = 109)	After (n = 90)	$X^{2}$
1. VHVs are surveyed on the 25th of each month. They are supposed to send data to the zone leader on the 28th of each month, to head of the village on the 30th, to the PCU for analysis, and to report to all stakeholders to prepare a dengue solution program. This process is called the larval indices surveillance system.		29 (25.7)	84 (74.3)	89.45 ***
2. The objective of the VHVs’ larval survey is to reduce dengue outbreaks.		93 (51.1)	89 (48.9)	11.62 ***
3. Larval indices formula = $\frac{\mathrm{No}\;\mathrm{of}\;\mathrm{positive}\;\mathrm{containers}}{\mathrm{No}\;\mathrm{of}\;\mathrm{houses}\;\mathrm{inspected}}\times 100$ Breteau index (BI)		26 (22.8)	88 (77.2)	110.10 ***
4. Larval indices formula = $\frac{\mathrm{No}\;\mathrm{of}\;\mathrm{positive}\;\mathrm{containers}}{\mathrm{No}\;\mathrm{of}\;\mathrm{houses}\;\mathrm{inspected}}\times 100$ House index (HI)		19 (18.8)	82 (81.2)	107.07 ***
5. Larval indices formula = $\frac{\mathrm{No}\;\mathrm{of}\;\mathrm{positive}\;\mathrm{containers}}{\mathrm{No}\;\mathrm{of}\;\mathrm{houses}\;\mathrm{inspected}}\times 100$ Container index (CI)		14 (14.9)	80 (85.1)	114.38 ***
6. The standard number for positive containers per 100 houses inspected is <50		7 (7.4)	88 (92.6)	164.91 ***
7. The standard level for percentage of houses infested with larva and/or pupae is <10		34 (29.1)	83 (70.9)	75.79 ***
8. The standard percentage of water-holding infested containers with larva is <1		12 (12.1)	87 (87.9)	144.68 ***
9. If the survey finds that among 20 houses there are 4 that are infested or that among 1000 water containers there are 200 that are infested, BI = 1000.		0 (0)	87 (100)	187.21 ***^, a^
10. If the survey finds that among 20 houses there are 4 that are infested or that among 1000 water containers there are 200 that are infested, HI = 20.		36 (31.6)	78 (68.4)	57.96 ***
11. If the survey finds that among 20 houses there are 4 that are infested or that among 1000 water containers there are 200 that are infested, CI = 20.		25 (23.4)	82 (76.6)	5.03 *
12. If the water container capacity is 100 L, we can use red lime in the container.		6 (6.3)	89 (93.7)	172.31 ***
13. The larval indices survey needs to be conducted every 7 days because the life cycle of *Aedes aegypti* ranges from 7 to 11 days.		5 (5.6)	84 (94.4)	157.05 ***
14. The results of the larval indices surveillance system must be documented every 25th of the month in the “violet book”.		27 (23.3)	89 (76.7)	111.39 ***
15. VHVs who are community leaders should collect data from zone leaders and send to PCUs every 30th of the month.		12 (11.9)	89 (88.1)	152.32 ***
**Total**	Poor level (cut-off point < 12)	109 (97.3)	3 (2.7)	0.000 ***^, a^
	Good level (cut-off point ≥ 12)	0 (0.0)	87 (100)	

VHV = village health volunteer; UDS = understanding dengue solution; $X^{2}$ = chi-square test. ^a^ Fisher’s exact test. * *p* < 0.05, *** *p* < 0.001.

**Table 6 ijerph-19-11989-t006:** Comparison of correct response rates regarding the UDS for family leaders before and after implementation of the training programs.

Family Leaders’ UDS		Correct Responses n (%)		$X^{2}$
		Before (n = 932)	After (n = 856)	
1. If a patient has a high fever for 2 to 7 days, petechiae, and a painful enlargement of the liver, the patient is showing signs of dengue infection.		794 (49.3)	817 (50.7)	52.57 ***
2. If the patient presents signs and symptoms of dengue, then the patient is showing signs of a dengue viral infection.		586 (44.5)	731 (55.5)	116.64 ***
3. If a person living in a high dengue risk area is infected with one dengue serotype, they may have lifelong immunity to that strain. However, they would still be vulnerable to other serotypes and could, thus, be infected with the other dengue serotypes later in life.		383 (37.6)	635 (62.4)	199.23 ***
4. If a patient is protected from female *Aedes aegypti* bites, they will be safe from dengue.		715 (48.2)	768 (51.8)	53.32 ***
5. If a patient has a high fever for 2–7 days, nausea, vomiting, and possible abdominal pain, the patient is in the fever stage.		555 (45.8)	657 (54.2)	60.47 ***
6. If a patient with dengue hemorrhagic fever presents pain at the right lower costal margin, they are showing signs of hepatomegaly.		442 (38.4)	709 (61.6)	234.84 ***
7. If a patient with dengue hemorrhagic fever presents signs of shock from leakage of plasma, they will have poor tissue perfusion, weak pulse, and narrowed pulse pressure.		648 (47.4)	719 (52.6)	51.88 ***
8. If your neighbor presents signs of poor tissue perfusion, weak pulse, and clammy skin, you need to send them to hospital.		720 (49.5)	735 (50.5)	21.83 ***
9. Dengue patients should avoid consuming aspirin or non-steroidal anti-inflammatory drugs because they may cause gastritis and subsequent massive gastrointestinal or hepatic injury.		331 (38.4)	530 (61.6)	124.57 ***
10. If your neighbor presents high fever on day 1, you give one paracetamol every 6 h and a tepid sponge bath.		585 (47.2)	655 (52.8)	39.69 ***
**Total**	Poor level (cut-off point < 8)	628 (70.4)	268 (29.6)	0.000 ***
	Good level (cut-off point ≥ 8)	304 (33.9)	592 (66.1)	

VHV = village health volunteer; UDS = understanding dengue solution; $X^{2}$ = chi-square test. *** *p* < 0.001.

**Table 7 ijerph-19-11989-t007:** Comparison of correct response rates regarding the ULISS for family leaders before and after implementation of the training programs.

Family Leaders’ ULISS		Correct Response n (%)		$X^{2}$
		Before (n = 932)	After (n = 856)	
1. The number of larvae of female *Aedes Aegypti* in the area is indicated by larval indices.		386 (42.3)	527 (57.7)	72.49 ***
2. Through the container index (CI), the value for identifying a dengue outbreak is calculated. Then, the number of water containers and the number of water containers infested with larvae are surveyed.		274 (40.8)	398 (59.2)	55.59 ***
3. The participation of family leaders in the survey for larval indices in and out of the household every 7 days is key.		546 (44.5)	680 (55.5)	90.05 ***
4. Insecticide Temephos sand granulates may be used to eliminate larvae but not to eliminate mosquito eggs.		41 (9.6)	388 (90.4)	409.85 ***
5. You suggest the use of lotion in the neighborhood as a prevention of mosquito bites.		465 (41.5)	656 (58.5)	136.44 ***
6. If people in the village have dengue, you suggest for all to prevent dengue by eliminating mosquito breeding sites and endeavoring to diminish the scores for the larval indices around the house.		739 (49.6)	752 (50.4)	23.60 ***
7. You suggest eliminating mosquito larvae by cleansing and scrubbing the edge of the container over the area that used to have water.		192 (30.8)	431 (69.2)	173.94 ***
8. You suggest putting the water container upside down until the larvae can be seen. You also check the container again, considering that the lifespan of a mosquito egg is 1–5 years.		195 (27.3)	520 (72.7)	294.89 ***
9. If the water container capacity is 100 L, we can use red lime in the container.		180 (23.9)	574 (76.1)	417.04 ***
10. The larval indices surveillance system must be documented on the 25th of every month in the “Violet book”.		548 (42.8)	732 (57.2)	157.79 ***
**Total**	Poor level (cut-off point < 8)	916 (64.2)	511 (35.8)	0.000 ***
	Good level (cut-off point ≥ 8)	16 (4.4)	345 (95.6)	

ULISS = understanding larval indices surveillance system; $X^{2}$ = chi-square test. *** *p* < 0.001.

**Table 8 ijerph-19-11989-t008:** Comparison of correct response rates regarding the UDS of student leaders before and after implementation of the training programs.

Student Leaders’ UDS		Correct Response n (%)		$X^{2}$
		Before (n = 59)	After (n = 59)	
1. If a patient has a high fever for 2 to 7 days, petechiae, and a painful enlargement of the liver, the patient is showing signs of dengue infection.		54 (48.6)	57 (51.4)	1.37 ^ns^
2. If the patient presents signs and symptoms of dengue, then the patient is showing signs of a dengue viral infection.		50 (49.5)	51 (50.5)	0.07 ^ns^
3. If a person living in a high dengue risk area is infected with one dengue serotype, they may have lifelong immunity to that strain. However, they would still be vulnerable to other serotypes and could, thus, be infected with the other dengue serotypes later in life.		32 (38.6)	51 (61.4)	14.66 ***
4. If a patient is protected from female *Aedes aegypti* bites, they will be safe from dengue.		58 (50.0)	58 (50.0)	0.00 ^ns^
5. If a patient has a high fever for 2–7 days, nausea, vomiting, and possible abdominal pain, the patient is in the fever stage.		35 (42.7)	47 (57.3)	5.76 *
6. If a patient with dengue hemorrhagic fever presents pain at the right lower costal margin, they are showing signs of hepatomegaly.		32 (38.1)	52 (61.9)	16.53 ***
7. If a patient with dengue hemorrhagic fever presents signs of shock from leakage of plasma, they will have poor tissue perfusion, weak pulse, and narrowed pulse pressure.		35 (40.2)	52 (59.8)	12.64 ***
8. If your neighbor presents signs of poor tissue perfusion, weak pulse, and clammy skin, you need to send them to hospital.		44 (44.0)	56 (56.0)	9.44 **
9. Dengue patients should avoid consuming aspirin or non-steroidal anti-inflammatory drugs because they may cause gastritis and subsequent massive gastrointestinal or hepatic injury.		16 (26.7)	44 (73.3)	26.58 ***
10. If your neighbor presents high fever on day 1, you give one paracetamol every 6 hr and a tepid sponge bath.		24 (33.8)	47 (66.2)	18.71 ***
**Total**	Poor level (cut-off point < 8)	45 (77.6)	13 (22.4)	0.000 ***
	Good level (cut-off point ≥ 8)	14 (23.3)	46 (76.7)	

UDS = understanding dengue solution; $X^{2}$ = chi-square test; ^ns^ not significant; * *p* < 0.05. ** *p* < 0.01. *** *p* < 0.001.

**Table 9 ijerph-19-11989-t009:** Comparison of correct response rates regarding the ULISS of student leaders before and after implementation of the training programs.

Student Leaders’ ULISS		Correct Response n (%) (n = 59)		$X^{2}$
		Before	After	
1. The number of larvae of female *Aedes aegypti* in the areas are indicated by the larval indices.		24 (44.4)	30 (55.6)	1.23 ^ns^
2. Through the container index (CI), the value for identifying a dengue outbreak is calculated. Then, the number of water containers and the number of water containers infested with larvae are surveyed.		18 (37.5)	30 (62.5)	5.06 *
3. The participation of family leaders in the survey for larval indices in and out of the household every 7 days is key.		20 (26.0)	57 (74.0)	51.17 ***
4. Insecticide temephos sand granulates may be used to eliminate larvae, but not to eliminate mosquito eggs.		8 (17.8)	37 (82.2)	30.21 ***
5. You suggest the use of lotion in the neighborhood as a prevention of mosquito bites.		31 (38.8)	49 (61.3)	12.58 ***
6. If people in the village have dengue, you suggest for all to prevent dengue by eliminating mosquito breeding sites and endeavoring to diminish the scores for the larval indices around the house.		44 (47.8)	48 (52.2)	0.79 ^ns^
7. You suggest eliminating mosquito larvae by cleansing and scrubbing the edge of the container over the area that used to have water.		3 (6.0)	47 (94.0)	67.19 ***
8. You suggest putting the water container upside down until the larvae can be seen. You also check the container again, considering that the lifespan of a mosquito is 1–5 years.		7 (13.2)	46 (86.8)	52.10 ***
9. If the water container capacity is 100 L, we can use red lime in the container.		4 (6.8)	55 (93.2)	88.17 ***
10. The larval indices surveillance system must be documented on the 25th of every month in the “Violet book”.		20 (27.4)	53 (72.6)	39.12 ***
**Total**	Poor level (cut-off point < 8)	59 (73.8)	21 (26.3)	0.000 ***^, a^
	Good level (cut-off point ≥ 8)	0 (0.0)	38 (100.0)	

ULISS = understanding larval indices surveillance system; $X^{2}$ = chi-square test. ^a^ Fisher’s exact test. ^ns^ Not significant. * *p* < 0.05. *** *p* < 0.001.

**Table 10 ijerph-19-11989-t010:** Risk of dengue in the village scores, list of stakeholders, larval indices (BI, HI, and CI) before and after implementation.

Village	RDV Scores	Total Stakeholders					Before (June 2020)			After (August 2020)		
		Household (N = 2372)	People (N = 7068)	VHVs (N = 134)	Head of Zone (N = 30)	Head of Village (N = 10)						
							BI	HI	CI	BI	HI	CI
V_1_	15 *	316	891	11	2	1	54.21	24.77	7.93	14.88	0.93	2.15
V_2_	14 *	287	795	12	3	1	82.45	40.43	8.47	30.27	24.32	3.71
V_3_	20 **	253	751	18	3	1	37.18	41.67	11.94	53.21	28.85	6.06
V_4_	20 **	312	1069	19	4	1	154.37	49.5	15.27	52.94	22.17	5.63
V_5_	14 *	294	715	15	5	1	112.46	48.15	12.93	25.82	1.1	3.61
V_6_	20 **	233	756	16	4	1	49.71	25.73	6.15	8.77	21.64	1
V_7_	16 *	195	673	12	2	1	340.13	77.71	25.09	85.44	40.51	6.59
V_8_	18 **	154	482	13	3	1	70.43	34.78	5.93	3.48	3.48	0.35
V_9_	12 *	168	442	9	2	1	103.42	41.88	8.28	29.06	23.93	2.8
V_10_	12 *	160	494	9	2	1	134.21	51.75	13.73	39.66	20.69	4.68

RDV = dengue risk village; BI = Breteau index: number of positive containers per 100 houses inspected (BI < 50); HI = house index: percentage of houses infested with larvae (HI < 10); CI = container index: percentage of water-holding containers infested with larvae (CI < 1); * L-RDV; ** H-RDV.

## Data Availability

All datasets are available upon request to the corresponding authors.

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
