# Peer review of "Effects of the Developing and Using a Model to Predict Dengue Risk Villages Based on Subdistrict Administrative Organization in Southern Thailand"

_ijerph, 2022, doi:10.3390/ijerph191911989_

Round 1
Reviewer 1 Report
Dear Editors,
The study entitled "Effects of the developing and using a model to predict dengue risk villages based on subdistrict administrative organization in southern Thailand" addresses an important issue in dengue control. The manuscript focuses on popular participation as means to implement measures to control dengue, reduce larval indices and disease morbidity. With great participation of communities and stakeholders the article is successful in proposing actions and results.
Major issues
The manuscript however needs a review in the Methods section as it is unclear how the study was prepared, how the participants were enrolled for the study and the data collected. The manuscript presents a difficulty in clearly presenting how the study methodology was carried out and how this impacts the results. Also, there is an excess of acronyms that make the text confusing throughout the manuscript and not very clear for a reader that must stop the reading to discover the meaning of tens of acronyms. Diagrams showing the sampling process in Methods section could be a a good way to become the text clear for the readers.
Other issues
P2L98 - what are these standarts?
P3L115 - how were these key persons enrolled? how many houses were sampled? Authors should explicit here the sampling methodology providing information on how the houses were chosen, how many houses?, % of houses sampled.
P3L139 – Authors should better explain the Cornbach coefficient. The coefficient of 0.70 is acceptable for what?
P3L142 - information on the questionnaires should come first than its reliability study.
P3L147 - this paragraph is confused. I could not understand what it really means
authors should provide a schematic presentation of the methodology to clear for readers the steps of the study
P4L174 - Does the larval indices correspond to the houses/volunteers sampled? I mean, are they in the same unit of study?
P5L204 - this text should be on Methods section. Again, diagrams showing the steps of sampling process should be appropriated here
P5L206 - When reading the Results the Methods become clear, what means that adjustments are needed.
P5L208 – all percentages should be presented as like this (702/793; 88.5%) and not like (65.8%) – in order to inform the reader the number of individuals;
P5L206-216 – Mean values should be accompanied by standard deviation
P5L225-245 – these paragraphs should be inserted in Methods section, because no results are presented here
P5L247-onwards- I think these paragraphs should be on Methods since dengue risk prediction is explained here
P6L254 – “the dengue herd immunity factor, which was demonstrable within the community and defined by the dengue morbidity rate of each village; “ – Why dengue herd immunity is related to morbidity rate?
P6L256 – “the current morbidity rate factor, referring to the value comparing the current dengue morbidity rates with the median rates of the past five years.” – Is the median of the mortality rate an appropriated epidemiological measure?
P6278 – why the cut-off was set on 17?
Table 1 - maybe due to the excess of acronyms in the text, resulting in reading difficulty, I could not understand how these values in table 1 were estimated - I imagine that some readers must experience the same difficulty
Figure 1 – authors should rewrite the legend providing more information about the figure
Table 2 - it’s not clear how the actions aiming understanding the disease and the LISS for volunteers and Stakeholders is linked to predicting the risk of dengue in the region
Tables 4 to 8 present many similar information considering volunteers, family leaders, students – authors should consider gathering all the information in one or two table, since many variables are exactly the same.
Table 9 – a statistical test should be provided as a way to prove the reduction of larval indices in the 10 villages after the implementation of the actions. / a list of acronyms should be given below the table
P20L477 – isn’t the dengue mortality rate reduction in 2021 related to Covid-19? In many countries the displacement of health professionals for covid control caused an impact on epidemiological surveillance and data collection of other diseases like dengue fever.
Author Response
//
Response to reviewer 1
Dear Reviewer
Thank you so much for your suggestion, and give me opportunity to rewrite and make clearly methodology and the results.
Best Regards
The study entitled "Effects of the developing and using a model to predict dengue risk villages based on subdistrict administrative organization in southern Thailand" addresses an important issue in dengue control. The manuscript focuses on popular participation as means to implement measures to control dengue, reduce larval indices and disease morbidity. With great participation of communities and stakeholders the article is successful in proposing actions and results.
Major issues
The manuscript however needs a review in the Methods section as it is unclear how the study was prepared, how the participants were enrolled for the study and the data collected. The manuscript presents a difficulty in clearly presenting how the study methodology was carried out and how this impacts the results. Also, there is an excess of acronyms that make the text confusing throughout the manuscript and not very clear for a reader that must stop the reading to discover the meaning of tens of acronyms. Diagrams showing the sampling process in Methods section could be a a good way to become the text clear for the readers.
Answer: Thank you so much for your kindness. According to major issues, we agree with you to rewrite only key abbreviations for easy reading, and draw the table 1 for present the participant in table 1 (P3L136).
We rewrite the methodology is five steps (P3L111-322) and separated the results following the objectives. (See the results P7L324).
Other issues
Q1: P2L98 - what are these standards?
Answer: we add the standards of dengue morbidity rate (50 cases/100000 populations) and mortality rate 0.2 percentages of 100 dengue patients [20]. (See P3L104)
Reference:
- Thai Ministry of Public Health. Manual of assessment district for sustainable disease control. Nontaburi Province: Department of Disease Control, Ministry of Public Health, 2013. http://data.ptho.moph.go.th/cdc/files/news/f01_20121219085230_93010000.pdf. (in Thai)
Q2: P3L115 - how were these key persons enrolled? how many houses were sampled? Authors should explicit here the sampling methodology providing information on how the houses were chosen, how many houses?, % of houses sampled.
Answer: Draw table 1 to show the amount of key participant (P3L136) and analysis of simple size of family leader by G* star power (P3L121)
Q3: P3L139 – Authors should better explain the Cornbach coefficient. The coefficient of 0.70 is acceptable for what?
Answer: According to reliability refers to the degree to which the results obtained by a measurement and procedure can be replicated.
Reference:
- Bolarinwa, O. Principles and methods of validity and reliability testing of questionnaires used in social and health science researches. Nigerian Postgraduate Medical Journal. 2015, 22(4), 195–201.
Q4: P3L142 - information on the questionnaires should come first than its reliability study.
Answer: Rewrite on topic UDS and ULISS assessment item before testing reliability (See P6L270)
Q5: P3L147 - this paragraph is confused. I could not understand what it really means
authors should provide a schematic presentation of the methodology to clear for readers the steps of the study
Answer: Rewrite the section of methodology as Materials and Methods in five step in order to develop Keawsan SAO dengue model consisted of five steps of CPAR (P4L111-P7L324).
Q6: P4L174 - Does the larval indices correspond to the houses/volunteers sampled? I mean, are they in the same unit of study?
Answer: it is the same group of family leaders who respond to UDS and larval indices survey of their houses. (See in table 1 P3L136)
Q7: P5L204 - this text should be on Methods section. Again, diagrams showing the steps of sampling process should be appropriated here
Answer: Rewrite the method of study followed suggest ion of the reviewer and identify in each step of CPAR (See P3L111-P7L324)
Q8: P5L206 - When reading the Results the Methods become clear, what means that adjustments are needed.
Answer: Thanks for your point, we rewrite the results of study divided into number 3 (P7L325-P21L720).
Q9: P5L208 – all percentages should be presented as like this (702/793; 88.5%) and not like (65.8%) – in order to inform the reader the number of individuals;
Answer: We change the word following that suggestion. It show in the result 3.1.2 houses’ environment (P8L339)
Q10: P5L206-216 – Mean values should be accompanied by standard deviation
Answer: We change the word following that suggestion. It show in the result 3.1.2 houses’ environment (P9L350)
Q11: P5L225-245 – these paragraphs should be inserted in Methods section, because no results are presented here
Answer: Move the paragraph to method of study. (SeeP6L240-262
Q12: P5L247-onwards- I think these paragraphs should be on Methods since dengue risk prediction is explained here
Answer: move the paragraph to method of study. (See P5L198-237)
Q13: P6L254 – “the dengue herd immunity factor, which was demonstrable within the community and defined by the dengue morbidity rate of each village; “ – Why dengue herd immunity is related to morbidity rate?
Answer: We add the meaning at P5L208-209 and attach supplement 1
“According to the natural course of dengue infection, the immune system is the body's primary defence against the virus. When someone is infected with dengue, the innate and adaptive immune responses together fight the virus. The B cells produce antibodies that specifically recognise and neutralise the foreign viral particles, and cytotoxic T cells recognise and kill infected cells with dengue virus. People who are infected subsequently with a different dengue virus type may experience "antibody-dependent enhancement," a condition that occurs when the immune response worsens dengue clinical symptoms, increasing the risk of severe dengue (1). (Supplement 1)”
Reference
1.Tang B, Xiao Y, Sander B, Kulkarni MA, RADAM-LAC Research Team, Wu J. Modelling the impact of antibody-dependent enhancement on disease severity of Zika virus and dengue virus sequential and co-infection. R Soc Open Sci 2020;7. (see supplement 1)
Q14: P6L256 – “the current morbidity rate factor, referring to the value comparing the current dengue morbidity rates with the median rates of the past five years.” – Is the median of the mortality rate an appropriated epidemiological measure?
Answer: According to the dengue outbreak based on dengue virus serotypes and the immunology in the risk area, with a low dengue incidence rate, in the following year, the area is at high-risk of an outbreak. When the current morbidity rate is less than the past 5-year median morbidity rate, the area is at high-risk of dengue outbreak. (See supplement 1)
Q15: P6278 – why the cut-off was set on 17?
Answer: Rewrite P5L233-237
“The risk cut-off value was set at 17 points. If a village scored 17 or more points, it was an H-RDV; if less than 17, an L-RDV based on the two level of risk that confirmed from a previous study, it can be practically understand high and low [11,12].”
References
- Suwanbamrung, C., Developing the active larval indices surveillance system for dengue solution in low and high dengue risk primary care units, Southern Thailand. J Health Res. 2018, 32(6), 408-420.
- Suwanbamrung, C.; Le, C.N.; Kaewsawat, S.; Chutipattana,N.; Khammaneechan, P.; Thongchan,S.; Nontapet, O.; Thongsuk, C.; Laopram, S.; Niyomchit, C.; Sinthu, R. Developing risk assessment criteria and predicting high- and low-dengue risk villages for strengthening dengue prevention activities: Community participatory action research, Thailand. J Prim Care Community Health. 2021, 12, 21501327211013298.
Q16: Table 1 - maybe due to the excess of acronyms in the text, resulting in reading difficulty, I could not understand how these values in table 1 were estimated - I imagine that some readers must experience the same difficulty
Answer: Thank you for your suggestion, we rewrite some acronyms. The details of criteria showed in supplement 1 and we add up reference number 12
Q17: Figure 1 – authors should rewrite the legend providing more information about the figure
Answer: We rewrite the six steps of the system (P10L374-397)
Q18: Table 2 - it’s not clear how the actions aiming understanding the disease and the LISS for volunteers and Stakeholders is linked to predicting the risk of dengue in the region
Answer: change Table 2 to Table 3, the risk dengue village data can encourage stakeholders of each village and score of risk dengue village come from activities. The village can see the scores and improved the activities as show 5 following the table (P9L369)
Q19: Tables 4 to 8 present many similar information considering volunteers, family leaders, students – authors should consider gathering all the information in one or two table, since many variables are exactly the same.
Answer: it is the same content of family leaders and student leaders, and we don’ t need to compare between groups. Then, it is not show in the same table.
Q20: Table 9 – a statistical test should be provided as a way to prove the reduction of larval indices in the 10 villages after the implementation of the actions. / a list of acronyms should be given below the table
Answer: we change table number 9 to table number 10 and rewrite the acronyms below table, So Thanks (P20L629-646)
Q21: P20L477 – isn’t the dengue mortality rate reduction in 2021 related to Covid-19? In many countries the displacement of health professionals for covid control caused an impact on epidemiological surveillance and data collection of other diseases like dengue fever.
Answer: Change the page to P21L673, the lower morbidity rate in 2021 can show data in the two primary care units, and this time just start of COVID-19 pandemic.

Reviewer 2 Report
The research Effects of the developing and using a model to predict dengue riskvillages based on subdistrict administrative organization in southern Thailand is very interesting and gives new information.
Comments on the paper:
Title reflects the paper’s content.
Abstract is appropriate.
Introduction
The introduction makes a proper introduction to the subject matter of the paper.
Materials and Methods, Results and Discussion
Well written.
Editoral comments:
Key words shold be dengue risk; village; larval indices; surveillance system; community participatory action research; subdistrict administrative organization
Line 64 should be spectively [7}.
Line 79 should be Alvarado-Castro et al. [18].
Lines 80-81 should be Traditionally, larval indices surveillance systems (LISS)
Table 8 should be e Aedes aegypti
Author Response
//
Reviewer 2
Dear Reviewer
We would like to say thank you very much for your kindness. We rewrite following your suggestion and integrated another reviewer and editor.
Best regards
Comments on the paper:
Title reflects the paper’s content.
Abstract is appropriate.
Introduction
The introduction makes a proper introduction to the subject matter of the paper.
Materials and Methods, Results and Discussion
Well written.
Editoral comments: (answer on Green color)
Q1: Key words should be dengue risk; village; larval indices; surveillance system; community participatory action research; subdistrict administrative organization
Answer: So thank you rewrite as you mention. (See P1L49)
Q2: Line 64 should be spectively [7}.
Answer: Rewrite (See P2L68)
Q3: Line 79 should be Alvarado-Castro et al. [18].
Answer: So thank you, we will change following your suggestion.(See P2L83)
Q4: Lines 80-81 should be Traditionally, larval indices surveillance systems (LISS)
Answer: Thanks for your suggestion (See P2L84)
Table 8 should be e Aedes aegypti
Answer: Thanks for your suggestion (See P19L573)

Round 2
Reviewer 1 Report
Dear editor,
The authors have adjusted the text of the manuscript according to the suggestions raised in this review for a better understanding for the reader. All major and minor issues have been responded accordingly.
Author Response
Dear reviewer
Thank you so much for your suggestion regarding the English language and style are minor spell check, We check and found some missing and spell words (Blue color). However, the full manuscript was edited in the master's English language following the certificate.